# An Adversarial Benchmark for Fake News Detection Models

**Lorenzo Jaime Yu Flores[1], Yiding Hao[1]**

[1]Yale University
New Haven, Connecticut 06520
{lj.flores, yiding.hao}@yale.edu

## Abstract

With the proliferation of online misinformation, fake news detection has gained importance in the artificial intelligence community. In this paper, we propose an adversarial benchmark that tests the ability of fake news detectors to reason about real-world facts. We formulate adversarial attacks that target three aspects of "understanding": compositional semantics, lexical relations, and sensitivity to modifiers. We test our benchmark using BERT classifiers fine-tuned on the LIAR (Wang 2017) and Kaggle Fake-News datasets (UTK Machine Learning Club 2017), and show that both models fail to respond to changes in compositional and lexical meaning. Our results strengthen the need for such models to be used in conjunction with other fact checking methods.

## Introduction

As online media plays an increasingly impactful role in modern social and political movements, the ability to detect and halt the flow of misinformation has become the subject of substantial research in the artificial intelligence community. An important component of this research is the task of *fake news detection*—a natural language classification task in which a model must determine whether a news article is intentionally deceptive (Rubin, Chen, and Conroy 2015). Unfortunately, fake news detection is as challenging as it is important. In order to successfully distinguish fake news articles from genuine ones, a model must not only be proficient in natural language understanding, but also be able to incorporate world knowledge into its computation, including knowledge of current events.

The inherent difficulty of this task, as well as the social and political incentives that encourage development of methods for evading content filters, raises questions surrounding the robustness of fake news detectors against adversarially written articles. To that end, a number of studies, such as Zhou et al. (2019), Ali et al. (2021), and Koenders et al. (2021), have subjected fake news detectors to a battery of attacks. All three of these studies have been able to produce cleverly written fake news articles that evade detection.

This paper proposes an adversarial benchmark for fake news detection that is designed to target three aspects of a

model's "understanding": whether it has the ability to employ semantic composition, whether it incorporates world knowledge of political parties, and whether adverb intensity is employed as a signal of fake news. Our benchmark is based on the premise that an ideal fake news detector should base its classification on the semantic content of its input and its relation to real-world facts, and not on superficial features of the text. This means that models that are vulnerable to our attacks are likely to be overly reliant on heuristics relating to word choice while failing to extract substantive assertions made by the articles they are tested on.

To test our benchmark, we fine-tune BERT classifiers (Devlin et al. 2019) on the LIAR dataset (Wang 2017) and the Kaggle Fake-News dataset (UTK Machine Learning Club 2017) and subject them to our three adversarial attacks. Since BERT is pre-trained on a large corpus of books (Zhu et al. 2015) and Wikipedia articles, it is possible that a BERT-based fake news detector might contain world knowledge that could be leveraged for fake news detection. For the most part, this is not borne out by our results: we find that our models are vulnerable to two of our three attacks, suggesting that they lack the ability both to extract the content of an article and to compare this content to the knowledge provided by the pre-training corpus.

## Related Work

A number of authors have employed neural text models for fake news classification. These include deep diffusion networks (Zhang, Dong, and Yu 2020), recurrent and convolutional networks (Ruchansky, Seo, and Liu 2017; Yang et al. 2018; Nasir, Khan, and Varlamis 2021), and BERT-based models (Ding, Hu, and Chang 2020; Kaliyar, Goswami, and Narang 2021). Common benchmarks for fake news detection are the LIAR dataset (Wang 2017) and the Kaggle Fake-News dataset (UTK Machine Learning Club 2017). Ding, Hu, and Chang's (2020) BERT-based model achieved state of the art results on the LIAR dataset, while Kaliyar, Goswami, and Narang's (2021) FakeBERT architecture achieved state of the art results on the Kaggle Fake-News dataset.

On adversarial attacks for fake news detection, previous literature has shown that fake news detection models can be fooled by carefully tweaked input. Ali et al. (2021) and Koenders et al. (2021) applied a series of text based adver-

| Original Statement | Modified Statement |
|---|---|
| **Negation Attack** | |
| EU, Finland can help settlement of Syria conflict: Iran parliament speaker. | EU, Finland can not help settlement of Syria conflict: Iran parliament speaker. |
| Julian Assange ends the suspense: "the source of hacked emails is not Russia" | Julian Assange ends the suspense: "the source of hacked emails is Russia" |
| **Party Reversal Attack** | |
| John Kerry rejects suggestions of U.S. involvement in Turkey coup | Sarah Sanders rejects suggestions of U.S. involvement in Turkey coup |
| Donald Trump threatens to cancel Berkeley federal funds after riots shut down Milo event. | Elizabeth Warren threatens to cancel Berkeley federal funds after riots shut down Milo event. |
| **Adverb Intensity Attack** | |
| The western banking system is totally broken, totally insolvent and totally corrupt. | The western banking system is broken, insolvent and corrupt. |
| Trump nation absolutely rejects Mitt Romney for secretary of state pick. | Trump nation rejects Mitt Romney for secretary of state pick. |

Table 1: Adversarial examples generated by the negation attack, party reversal attack, and adverb intensity attack

sarial attacks including Text Bugger (Li et al. 2019), TextFooler (Jin et al. 2020), DeepWordBug (Gao et al. 2018) and Pruthi (Pruthi, Dhingra, and Lipton 2019). These are generic attacks for natural language models consisting of textual noise such as typos, character swaps, and synonym substitution. In addition to these standard attacks, Zhou et al. (2021) proposed three novel challenges for fake news detectors: (1) modifying details of a sentence involving time, location, etc., (2) swapping the subject and object of a sentence, and (3) adding causal relationships between events in a sentence or removing some of its parts.

The attacks we mention above mainly simulate noise that might appear in online text. In contrast, the attacks we propose are specifically tailored to the problem of fake news detection, particularly in the context of politics. Our attacks are not designed to simulate naturally occurring noise, but rather to test whether deep-learning models understand text, learn real-world facts, and employ inferential reasoning.

## Adversarial Attacks

For this paper, we consider a statement to be *fake* if it is factually incorrect, and *real* otherwise. We choose three attacks that would test a model's understanding of text and real-world facts. Our goal is to see whether the models tweak their outputs accordingly when the truthfulness of an input has been changed, or keep them unchanged otherwise. We

provide examples of each attack in Table 1.

For each adversarial attack, we input the original and modified statements into the model. Then, we compute (1) the percentage of instances where the predicted label was different for the original and modified statement ($\%_{\text{LabelFlip}}$), and (2) the average change in output probability that the statement is fake ($\Delta_{\text{Prob}}$), where a positive change means the attack increases the probability that the statement is fake.

### Negating Sentences

In the first attack, we negate the sentences of each input text using a script due to Bajena (2017). The script heuristically attempts to identify sentences with a third-person singular subject, and changes linking verbs such as *is*, *was*, or *should* into *is not*, *was not*, and *should not*, and *vice versa*. While the script is not guaranteed to negate a sentence completely, we assume that it tweaks the semantics of the dataset enough to justify a conspicuous effect on the classification probabilities. We assume that an ideal fake news detector would assign opposite labels to a text and its negation.

### Reversing Political Party Affiliations

In the second attack, we attempt to reverse the political party affiliations of named individuals appearing in the text. We identify names of American politicians in the text along with their party affiliations, and filter statements to those containing names from the Republican or Democratic Party. Then, we manually filter the remaining statements to only include real statements where replacing the original name with a random one would make the sentence untrue. In each of these texts, we replace names of Democrats with a randomly selected Republican, and *vice versa*.

The statements in the adversarial dataset consist of quotes, facts, or events associated with particular individuals. We therefore expect that name replacement should cause the model to classify a modified statement as fake.

### Reducing Intensity of Statements

In the third attack, we remove adverbs that increase sentences' intensity (e.g. *absolutely*, *completely*). We hypothesize that fake news is correlated with "clickbait" titles containing highly charged words (Alonso et al. 2021).

Removing polarizing words does not change the meaning of a sentence, thus the label should not change. For this attack, we input fake statements into the model, and expect that the model should still classify them as fake.

## Experimental Setup

We test our benchmark on three fine-tuned BERT$_{\text{BASE}}$ classifiers: two trained on the LIAR dataset and one trained on the Kaggle Fake-News dataset. For each benchmark, we apply our three transformations to the detector's test set, present the resulting texts to the appropriate models, and report the two metrics from the previous section, $\%_{\text{LabelFlip}}$ and $\Delta_{\text{Prob}}$.[1]

---

[1]The code for our experiments is available at the following repository: https://github.com/ljyflores/fake-news-explainability.

| Dataset | SOTA | Our Model |
|---|---|---|
| LIAR 2 Classes | — | **57.5** |
| LIAR 6 Classes | 27.3 | **29.4** |
| Kaggle Fake-News | **98.9** | 98.8 |

Table 2: Test set accuracy attained by our models, compared with previously reported state-of-the-art results.

## Models

Below we describe our three models.

**LIAR Models**  LIAR (Wang 2017) is a six-class dataset that classifies statements made by politicians as *True*, *Mostly True*, *Half True*, *Barely True*, *False*, and *Pants on Fire*. We train two models on this dataset, which differ in the number of possible output labels the model can predict. First, to verify that our BERT model achieves a level of performance comparable with the results reported by Ding, Hu, and Chang (2020) for LIAR, we train a six-class BERT classifier on the original version of the dataset. Next, in order to facilitate compatibility with the adversarial attacks, we train a two-class model that collapses the *True*, *Mostly True*, and *Half True* labels into a single *Real* class and the *Barely True*, *False*, and *Pants on Fire* labels into a single *Fake* class.

**Kaggle Fake-News Model**  The Kaggle Fake-News dataset (UTK Machine Learning Club 2017) is a two-class dataset consisting of headlines and text from news articles published during the 2016 United States presidential election. Our third model is a two-class classifier fine-tuned on this dataset. Since the officially published version of the dataset only contains gold-standard labels for the training data, we use 70% of the training set for training and the remaining 30% for testing.

## Feature Saliency Analysis

In addition to reporting $\%_{\text{LabelFlip}}$ and $\Delta_{\text{Prob}}$, we compute saliency maps for our Kaggle Fake-News model using the Gradient $\times$ Input method (G $\times$ I, Shrikumar, Greenside, and Kundaje 2017; Shrikumar et al. 2017) to measure how individual words impact the models' classifications. G $\times$ I is a local explanation method that quantifies how much each input contributes to the output logits. In G $\times$ I, the contribution of a feature is measured by the value of its corresponding term in a linear approximation of the target output unit. We obtain token-level saliency scores by adding together the saliency scores assigned to the embedding dimensions for each token.

## Results

Before discussing our results, we validate the quality of our models by comparing their performance with the current state of the art. These results are shown in Table 2. The six-class version of our LIAR model slightly outperforms the BERT-Based Mental Model of Ding, Hu, and Chang (2020), while our Kaggle Fake-News model achieves a compara-

| Dataset | $\%_{\text{LabelFlip}}$ | $\Delta_{\text{Prob}}$ |
|---|---|---|
| LIAR 2 Classes | 15.5 | 0.021 |
| Kaggle Fake-News | 0.3 | −0.0001 |

Table 3: Impact of the negation attack on our models.

| Dataset | $\%_{\text{LabelFlip}}$ | $\Delta_{\text{Prob}}$ |
|---|---|---|
| LIAR 2 Classes | 20.0 | 0.052 |
| Kaggle Fake-News | 4.0 | 0.014 |

Table 4: Impact of the political party reversal attack on our models.

ble level of performance to Kaliyar, Goswami, and Narang's (2021) FakeBERT model.[2]

## Negation Attack

Table 3 shows the impact of the sentence negation adversarial attack on the outputs of our two-class models. The Kaggle Fake-News model proves to be much more vulnerable to this attack than the LIAR model, though the vast majority of predictions were unchanged for both models. We observe in particular that negation causes only in a small increase in the probability scores assigned to the *Fake* class, despite the fact that the negation script targets the main auxiliary verb of the sentence, which typically has the effect of completely reverses the meaning of a sentence.

## Party Reversal Attack

Table 4 shows the impact of the name replacement attack on the models. Again, we find that the Kaggle Fake-News model is more susceptible to this attack than the LIAR model. Although most labels are still unchanged, we find that this attack has a greater impact on our models than the negation attack. It is therefore likely that our models are more sensitive to lexical relationships between specific words appearing in a statement than to the syntactic relationships that govern negation.

## Adverb Intensity Attack

Table 5 shows the impact of the intensity-reduction attack on the models. As shown, this attack has almost no effect on the models' output. Since the expected behavior is for the output predictions to remain unchanged, our models can be deemed to be robust to this attack. This result suggests that adverb intensity is not a significant heuristic for fake news classification.

---

[2]It is worth noting that Kaliyar, Goswami, and Narang (2021) did not perform a train–test split on the officially published training data for Kaggle Fake-News, but instead used the entire training set for both training and evaluation. Thus, the SOTA result in Table 2 is not directly comparable with our result, since the former may be inflated due to overfitting.

| Dataset | %$_{\text{LabelFlip}}$ | $\Delta_{\text{Prob}}$ |
|---|---|---|
| LIAR 2 Classes | 0.0 | 0.027 |
| Kaggle Fake-News | 0.9 | −0.008 |

Table 5: Impact of the adverb intensity attack on our models.

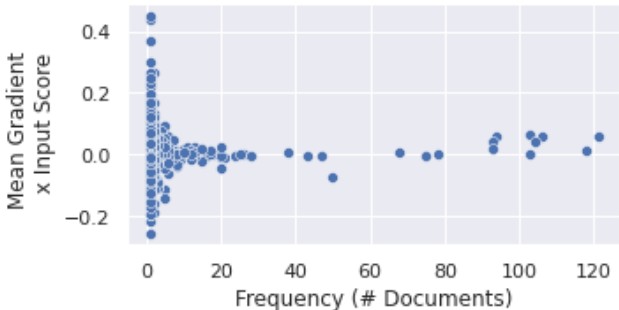

Figure 1: On average, words that appear more frequently in the datasets are assigned saliency scores closer to 0.

## Saliency Analysis

We use G × I heatmaps to identify keywords that may serve as signals for one class over the other. Due to its superior test set performance, we apply the saliency analysis to our Kaggle Fake-News model. We compute saliency scores for the *Fake* class, so that a positive saliency score means that a word increases the likelihood that the input is fake.

Figure 1 shows that frequency affects the degree to which a word may be associated with real or fake statements. Here, we find that words which appear in fewer documents are assigned more extreme saliency scores. Among the top 30 words with the most extreme G × I scores are names that appear once or twice in the dataset, such as *Sanford*, *Jody*, *Marco*, and *Gore*. In contrast, frequently-occurring names such as *Trump*, *Hillary*, and *Obama* have average G × I scores close to zero. This is likely because frequently-occurring names appear in a wider variety of publications, preventing them from being consistently associated with any particular ideological bias.

Figure 2 visualizes the impact of high-intensity adverbs on our model. Observe that the adverbs *totally* and *completely* have small G × I scores in comparison to other words in the sentence. This reflects the resilience of our model against the adverb intensity attack.

## Conclusion

In this study, we have created an adversarial benchmark for fake news detection that is designed to test models' ability to reason about real-world facts. We find that our BERT-based models are vulnerable to negation and party reversal attacks, whereas they are robust to the adverb intensity attack. For all three attacks, our model did not change its prediction in the vast majority of cases, and accordingly the only attack our models were robust to was the one that required the models' behavior to remain unchanged. It may be the case that

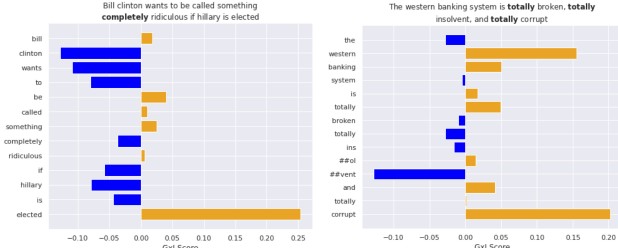

Figure 2: High-intensity adverbs have relatively small contributions to the output logits.

the models are simply unresponsive to the perturbations we performed on the inputs.

Deep learning has demonstrated an impressive level of competence in learning dependencies and relationships in natural language tasks. However, our findings suggest that current techniques are still not sufficient for tasks like fake news detection that require sophisticated forms of reasoning. As the state of the art in fake news detection continues to advance, our benchmark will serve as a valuable metric for the reasoning capabilities of future models.

These findings strengthen the need for fake news classification models to be used in conjunction with other fact checking methods. Other work made strides in this area by exploring features like comments on an article (Shu et al. 2019) or article interaction metrics article (likes, shares, retweets) that may signify an article is being maliciously spread (Prakash and Tucker 2021; Tschiatschek et al. 2018), or the possibility of incorporating crowd sourced knowledge or human fact checkers into the process altogether (Demartini, Mizzaro, and Spina 2020; Pennycook and Rand 2019).

We also observed that the model trained on LIAR was more sensitive (i.e. more labels were flipped) than the model trained on the Fake-News dataset. Upon reading the data, we observed that statements in LIAR were generally less polar and more focused on facts, whereas the Fake-News dataset appeared to be a mixed bag of headlines with more polarizing words. This suggests that data quality greatly impacts models' ability to learn facts and understand text.

Limitations of this work are that (1) the models were trained on only two datasets, and the results may not generalize to statements unrelated to general US politics, (2) computational limitations only let us explore shallow neural network architectures, and (3) the adversarial attacks we tried were relatively simple, and a real human may be able to negate or change the intensity of a sentence in more complex ways. Future work could employ more data sets as the training corpus, explore deeper model architectures, and use more complex adversarial attacks, for a more robust evaluation of these fake news models.

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
