# OpenReview forum: "An Adversarial Benchmark for Fake News Detection Models"
_AAAI.org/2022/Workshop/AdvML — AAAI-22 AdvML Workshop ShortPaper_

### Official Review · Reviewer_3VAy · 2021-11-29
**An Adversarial Benchmark for Fake News Detection Models**

**Rating:** 6
**Confidence:** 3

**Review:**

This paper proposes an adversarial benchmark for fake news detection, which is designed to evaluate models’ ability to reason about real-world facts. And some findings can strengthen the need for fake news classification models.

However, the author is also expected to make more analysis and experiments for a benchmark. A clearer discussion and definition should be also considered.

---

### Decision · Program_Chairs · 2021-12-01

**Decision:**

Accept (Short Paper)

**Comment:**

The reviewer agrees to accept this paper. Please consider the reviewer's comment in the camera-ready version.